# Polarization-Sensitive Electro-Optic Sampling of Elliptically-Polarized Terahertz Pulses: Theoretical Description and Experimental Demonstration

**Kenichi Oguchi, Makoto Okano and Shinichi Watanabe *** 

Department of Physics, Faculty of Science and Technology, Keio University, 3-14-1, Hiyoshi, Kohoku-ku, Yokohama, Kanagawa 223-8522, Japan; oguchi@wlab.phys.keio.ac.jp (K.O.); okano@phys.keio.ac.jp (M.O.)

* Correspondence: watanabe@phys.keio.ac.jp; Tel. +81-045-566-1687

**Abstract:** We review our recent works on polarization-sensitive electro-optic (PS-EO) sampling, which is a method that allows us to measure elliptically-polarized terahertz time-domain waveforms without using wire-grid polarizers. Because of the phase mismatch between the employed probe pulse and the elliptically-polarized terahertz pulse that is to be analyzed, the probe pulse senses different terahertz electric-field (E-field) vectors during the propagation inside the EO crystal. To interpret the complex condition inside the EO crystal, we expressed the expected EO signal by "frequency-domain description" instead of relying on the conventional Pockels effect description. Using this approach, we derived two important conclusions: (i) the polarization state of each frequency component can be accurately measured, irrespective of the choice of the EO crystal because the relative amplitude and phase of the E-field of two mutually orthogonal directions are not affected by the phase mismatch; and, (ii) the time-domain waveform of the elliptically-polarized E-field vector can be retrieved by considering the phase mismatch, absorption, and the effect of the probe pulse width. We experimentally confirm the above two conclusions by using different EO crystals that are used for detection. This clarifies the validity of our theoretical analysis based on the frequency-domain description and the usefulness of PS-EO sampling.

**Keywords:** terahertz time-domain spectroscopy; electro-optic sampling; phase-matching condition; polarization analysis

## 1. Introduction

Terahertz time-domain spectroscopy (THz-TDS) is a powerful tool to obtain coherent electric-field (E-field) time-domain waveforms in the terahertz frequency range [1,2], and it is an emerging spectroscopic technique for determining the low-energy dielectric and conductive properties of materials. THz-TDS relies on measuring the temporal waveform of a terahertz E-field pulse by using probe laser pulses with pulse widths (typically ~100 fs) shorter than the oscillation period of the terahertz pulse (typically a few ps). One can obtain the terahertz E-field time-domain waveform by changing the time interval between the optical probe and terahertz pulses [3]. By performing the Fourier transform of the waveform, the frequency-dependent amplitude and phase of the terahertz E-field pulse can be simultaneously obtained without using the Kramers–Kronig relation [4]. The complex dielectric properties of a sample can then be obtained by comparing the terahertz pulse amplitudes and phases measured with and without sample. This type of coherent terahertz spectroscopy has dramatically changed spectroscopic researches; THz-TDS has already been widely applied to investigation of materials [5–9].

To date, several detection methods have been proposed for accurate THz-TDS measurements. The electro-optic (EO) sampling is one of the most popular detection methods. Using this method,

one can obtain the information of the terahertz E-field by measuring the terahertz-E-field-induced polarization change of the probe pulse inside the EO crystal via the Pockels effect [2,10]. The measurement bandwidth is determined by the velocity-matching condition between the terahertz and probe pulses inside the EO crystal. Although the application of the EO sampling method was initially restricted to the terahertz frequency range due to the optical properties of the EO crystals, broadband spectroscopy is nowadays achieved not only in terahertz [2,10–14], but also in mid-infrared [15–23] and near-infrared [24] regions. This progress was possible owing to the discovery of many classes of EO crystals, such as ZnTe [11–13], GaP [14], GaSe [16,18,19], GaAs [25], $BaTiO_3$ [26], $LiNbO_3$ [13], $LiTaO_3$ [13], trans-4′-(dimethylamino)-N-methyl-4-stilbazolium tosylate (DAST) [27–29], $AgGaS_2$ [22,23], $ZnGeP_2$ [30], and ZnSe [30], and by choosing the optimal EO crystal and its thickness. The extension of THz-TDS to an ultra-broadband spectrum coverage enables the investigation of a wide variety of interesting phenomena [23,31–35].

Another promising application area of broadband spectroscopy is the polarization spectroscopy [36–39]. Polarization-sensitive (PS) spectroscopy is useful for characterizing different material properties, such as birefringence [40–45], the Higgs mode in superconductors [46,47], and vibrational circular dichroism [48,49]. By combining THz-TDS with ellipsometry techniques, anisotropic complex refractive indices can be obtained accurately in the terahertz region [50–52] as well as the mid-infrared region [53]. For measuring the polarization-dependent properties, much effort has been devoted to the development of PS devices, such as polarizers [54–56] and polarization rotators [57–62], in the terahertz frequency range. For instance, sensitive polarimetry has been realized by rotating a quarter-wave plate (QWP) [63] or a polarizer [64,65]. Other PS measurements have been also realized by utilizing PS detectors, such as EO crystals [66–68], multiple-contact photo-conductive antennas [69–73], rotating antennas [74], air plasma biased electrodes [75,76], EO sampling with beam splitter and two balanced detectors [77,78], and EO sampling with photo-elastic [79] or even electro-optic [80] modulators.

Among the various polarization detection techniques that are described above, the utilization of the EO crystal for detection has some unique advantages over the other techniques. Firstly, the full polarization information of the terahertz wave can be simply obtained by rotating the EO crystal [66–68] without any wire-grid polarizers. Consequently, the measurement system based on the rotating EO crystal method is simpler and the frequency bandwidth of the system is not limited by that of the wire-grid polarizer. Secondly, the polarization sensitivity of the detection method employing the EO crystal is determined by the symmetry of the nonlinear susceptibility tensor, the so-called $d_{ijk}$, of the EO crystal [66–68]. This means that we can easily formulate a polarization measurement procedure according to the symmetry structure of $d_{ijk}$ of the utilized EO crystal. As a result, when one utilizes two EO crystals with same crystal symmetry, a single signal processing procedure is sufficient for analyzing the polarization information. Thirdly, because this setup does not rely on any artificial PS devices, it is not necessary to implement or even design such a device, making the polarization measurement more cost-effective.

Although there are clear advantages of implementing an EO crystal for detection in the polarization spectroscopy setup, the PS THz-TDS measurement based on the rotating EO crystal has some inherent problems. One of the most serious problems concerns the velocity matching [12,14,81–86]. In general, the probe pulse and the terahertz pulse propagate with different speeds inside the EO crystal, because of the frequency-dependent refractive index of the EO crystal. Thus, during the propagation inside the EO crystal, the probe pulse is subject to a complex polarization change that is induced by the changing amplitude and polarization orientation of the terahertz pulse. The polarization change of the probe pulse certainly reflects the terahertz E-field time-domain waveform, but the E-field time trace that is recorded by scanning the delay time between terahertz and probe pulses cannot be interpreted straightforwardly. To date, several researchers have tried to interpret the E-field time traces that were obtained by the EO sampling method and demonstrated how to retrieve the original terahertz time-domain E-field waveform by taking into account for the velocity-matching problem [12,14,81–86]. However, all previous

reports limit their interpretation to linearly-polarized terahertz pulses, because the situation inside the EO crystal is rather complicated in the case of elliptically-polarized terahertz waves.

When the terahertz wave is elliptically polarized, the velocity-mismatch problem becomes more complex. Because of the difference in the velocities of the elliptically-polarized terahertz wave and the probe pulse, the terahertz E-field direction that has a strong impact on the modification of the polarization state of the probe pulse, changes as the probe pulse propagates inside the EO crystal. This is in stark contrast to the previous reports regarding the linearly-polarized terahertz waves, where the terahertz E-field direction is constant.

To overcome the velocity-mismatch problem that is encountered for elliptically-polarized terahertz pulses, we adopted the frequency-domain description, which was proposed by Gallot and Grischkowsky [83] for the linearly-polarized terahertz pulse detection, instead of the conventional Pockels effect description. Using the frequency-domain parameters, the EO signal can be expressed by sum- and difference-frequency generation between the terahertz and probe E-fields. In this description, the effects of frequency-dependent phase-mismatch (corresponding to the velocity mismatch in the Pockels effect description), the absorption coefficient, the nonlinear optical coefficient, and the effect of the finite pulse width of the probe pulse are straightforwardly included. Because of these benefits, the frequency-domain description has been used to describe the EO signals in different types of EO sampling setups, such as single-shot [85,87–89], nonellipsometric noncollinear [90], heterodyne [91], energy-sensitive [92], and shot-noise reduced [22] EO samplings.

In this invited review, we summarize our recent works regarding the PS-EO sampling method which allows us to measure elliptically-polarized terahertz time-domain waveforms [93–95]. Instead of the conventional Pockels effect description, we expressed the obtained EO signal based on the frequency-domain description [83]. By using our developed E-field reconstruction protocol, we were able to determine the frequency-dependent polarization parameters, i.e., ellipticity angle and angle of rotation. We also succeeded in retrieving the original elliptically-polarized terahertz E-field time-domain waveform in front of the EO crystal. In Section 2, we formulate the model for the PS-EO sampling and we interpret the EO signal in terms of both conventional Pockels effect and frequency-domain descriptions. By comparing those two theoretical formulations, we prove the importance of the frequency-domain description for the accurate reconstruction of terahertz time-domain waveforms. In Section 3, we present the two experimental results that are based on the PS-EO sampling. First, we focus on the polarization spectroscopy experiments. From the experimental results, we clarify that the same polarization state of terahertz wave is obtained regardless of the chosen EO crystal. Afterwards, we focus on the retrieval of the elliptically-polarized terahertz time-domain waveform by compensating for the frequency-dependent optical properties. We confirmed that the correction by appropriately considering the phase mismatch is quite important for the retrieval, and the time-domain waveforms that were retrieved from EO signals recorded using different EO crystals show good agreement. These results strongly support the effectiveness of the frequency-domain description for the PS-EO sampling. Finally, in Section 4, we conclude our results.

## 2. Theory

In this section, we explain our theoretical description of the data obtained by the PS-EO sampling based on the rotation of the EO crystal. As mentioned in the previous section, when an elliptically-polarized terahertz wave is measured by the EO sampling method without any terahertz polarizers, the probe pulse polarization is affected by a changing magnitude and polarization orientation of the terahertz E-field vector as it propagates through the EO crystal. Since the velocity mismatch between the terahertz and the probe pulses induces a relatively complex situation, a thorough theoretical description is indispensable for the appropriate interpretation of the measured EO signal and reconstruction of the original terahertz time-domain waveforms.

In Section 2.1, we explain a primitive but intuitive picture of the PS-EO sampling based on the Pockels effect inside the EO crystal, including the velocity mismatch. In order to account for

the velocity mismatch, this description considers that the EO crystal comprises many thin layers, and the probe pulse is affected by a different refractive index ellipsoid in each layer as a result of the changing terahertz E-field vector (hereafter referred to as "multilayer model"). This description enables an intuitive interpretation of the measured E-field time trace under velocity-mismatch conditions. However, this description cannot include the effect of the finite pulse width of the probe laser pulse, resulting in a more or less inaccurate prediction of the original time-domain waveform. In Section 2.2, we introduce the more theoretically rigorous frequency-domain description based on the frequency mixing between the E-fields of the terahertz and probe pulses inside the EO crystal. The obtained time-domain signals at two specified EO crystal orientations are described as the mutually orthogonal frequency-domain components of the elliptically-polarized terahertz pulse modified by the phase mismatch, absorption losses, and finite pulse width of the probe laser pulse. The effect of the finite pulse width of the probe pulse, which is not considered in the multilayer model, is naturally included in the frequency-domain description. In Section 2.3, we discuss one of the important results that were derived in the framework of the frequency-domain description, even though the terahertz E-field vector time-domain waveform is distorted by the phase mismatch, absorption, and effects of the finite pulse width, the polarization information of each frequency component can be easily extracted from the experimental data. In Section 2.4, we explain the procedure that allows us to retrieve the original terahertz E-field vector time-domain waveform in front of the EO crystal.

*2.1. Multilayer Model*

In this section, we introduce the multilayer model to interpret the E-field time traces of elliptically-polarized terahertz pulses that are measured by the PS-EO sampling method, including a certain velocity mismatch.

First, we consider the simple case of velocity matching, i.e., when the phase velocity of the terahertz wave and the group velocity of the probe pulse are exactly the same. In the Pockels effect description, the measured EO signal can be described by the terahertz E-field-induced birefringence in the EO crystal. After the linearly-polarized probe pulse has passed through the EO crystal that is simultaneously exposed to terahertz radiation, the probe pulse is elliptically-polarized due to the terahertz E-field-induced birefringence of the EO crystal. If the velocity matching condition is satisfied, both the magnitude and direction of the terahertz E-field vector that overlaps with the probe pulse stays constant as the probe pulse propagates through the EO crystal, irrespective of the polarization state of the terahertz pulse. Therefore, the terahertz E-field-induced birefringence that governs the rotation of the probe pulse polarization direction is constant. Under this condition, the Jones vector of the probe pulse after passing the EO crystal, $\begin{pmatrix} E_X \\ E_Y \end{pmatrix}$, can be simply described by

$$\begin{pmatrix} E_X \\ E_Y \end{pmatrix} = R(-\alpha) \begin{pmatrix} \exp\left(-\frac{iC}{2}\right) & 0 \\ 0 & \exp\left(\frac{iC}{2}\right) \end{pmatrix} R(\alpha) \begin{pmatrix} 1 \\ 0 \end{pmatrix}, \tag{1}$$

where $\alpha$ is the angle between the *X*-axis (which is defined by the initial polarization direction of the probe pulse) and the slow optic axis of the terahertz E-field-induced birefringence, *R* is the rotation matrix defined as $\begin{pmatrix} \cos\alpha & -\sin\alpha \\ \sin\alpha & \cos\alpha \end{pmatrix}$, and *C* is the phase retardation between the slow and fast optic axes of the birefringent EO crystal, which is proportional to the magnitude of the terahertz E-field that overlaps with the probe pulse. The Z-axis defines the propagation direction of the pulses. In Equation (1), the polarization state of the probe pulse before impinging on the EO crystal is linearly polarized along the X-direction, which is equivalent with the Jones vector $\begin{pmatrix} 1 \\ 0 \end{pmatrix}$. When we assume

a weak terahertz E-field, i.e., $C \ll 1$, $\begin{pmatrix} E_X \\ E_Y \end{pmatrix} \cong \begin{pmatrix} 1 - iC\cos 2\alpha \\ -iC\sin 2\alpha \end{pmatrix}$ is obtained from Equation (1).
In the conventional EO sampling setup [96], one measures the intensity difference between the *X*- and *Y*-components of the probe pulse after it has passed through the EO crystal, a QWP, and a Wollaston prism by using a balanced photo-detector. In this balanced detection scheme, the intensity difference *S* is proportional to the Y-component of the probe field, i.e., $S \propto C\sin 2\alpha$ [97]. Since the slow optic axis of terahertz E-field-induced birefringence is determined by both the E-field vector direction and the EO crystal orientation [67], the angle $\alpha$ can be used to extract the information of the E-field vector direction by changing the EO crystal orientation [66–68].

Next, we consider a phase velocity of the terahertz wave that differs from the group velocity of the probe pulse, i.e., a finite velocity mismatch. Bakker et al. [81] considered this condition for a linearly-polarized terahertz E-field. In this case, the magnitude of the terahertz E-field that overlaps with the probe pulse changes as the probe pulse propagates through the EO crystal. Thus, the retardation *C* becomes a function of the position *Z* and time *t*, i.e., $C(Z,t)$. Here, we define the time delay between the terahertz wave and probe pulse as $\tau$, and *S* has to be expressed as a function of $\tau$. Because $C(Z,t)$ is proportional to the terahertz E-field, denoted by $E_1(Z,t)$, the intensity difference signal $S(\tau)$ (hereafter referred to as "EO signal") is a function of the $E_1(Z,t)$ that overlaps with the probe pulse. According to the formulation by Bakker et al. [81], $S(\tau)$ can be written as a double integration over position *Z* and time *t*,

$$S(\tau) \propto r_{41} \int_0^l dZ \int_{-\infty}^{+\infty} dt\, I_{\mathrm{OPT}}(Z, t - \tau) E_1(Z, t). \tag{2}$$

Here, $r_{41}$ is the electro-optic coefficient for zinc-blende crystals, *l* is the thickness of the EO crystal, and $I_{\mathrm{OPT}}(Z, t - \tau)$ is the intensity profile of the probe pulse. However, when the terahertz E-field is elliptically-polarized, the situation becomes much more complex. In this case, not only the magnitude but also the direction of the terahertz E-field vector overlapping with the probe pulse changes as the probe pulse propagates through the EO crystal. Because the slow-optic-axis orientation $\alpha$ depends on the direction of the terahertz E-field vector, the angle $\alpha$ also becomes a function of position *Z* and time *t*, i.e., $\alpha(Z,t)$. Therefore, the rotation matrix $R(\alpha)$ in Equation (1) also depends on these two parameters.

To interpret the detected EO signal for elliptically-polarized terahertz waves, we utilized the multilayer model [98]. Figure 1 shows a schematic illustration of the multilayer model. In this model, we divide the EO crystal with thickness *l* into *N* layers, where the number of layers is much larger than unity and the thickness of each layer $l/N$ is much smaller than the coherence length of the EO crystal. Therefore, we can consider that, in each extremely thin layer, the velocity-matching condition between the terahertz wave and probe pulse is satisfied. Because of the velocity matching, the probe pulse senses a constant terahertz E-field-induced birefringence during the propagation through a single layer. At the boundary between the *n*-th and (*n* + 1)-th layers, we add the effects of the velocity mismatch, because the probe pulse senses a terahertz E-field-induced birefringence that is different from that in the previous layer. Although there is refractive index change between the two layers, we neglect the reflections at the boundaries, because the difference of the refractive index between two adjacent layers is small. Strictly speaking, if the probe pulse has a finite pulse width (as described by the intensity profile $I_{\mathrm{OPT}}(Z,t)$), it simultaneously senses successive changes of the terahertz E-field-induced birefringence within its pulse width. In the present multilayer model, we neglect such a complicated situation and suppose that the probe pulse is a delta-function-like ultra-short pulse, i.e., $I_{\mathrm{OPT}}(Z,t) \propto \delta(t - Z/v_g)$, where $v_g$ is the group velocity of the probe pulse.

In the following we formulate the Jones vector of the probe pulse in the framework of the multilayer description. The Jones vector of the probe pulse after it has passed through the first layer of the EO crystal is given by

$$R(-\alpha_1)\begin{pmatrix} \exp\left(-\frac{iC_1}{2}\right) & 0 \\ 0 & \exp\left(\frac{iC_1}{2}\right) \end{pmatrix} R(\alpha_1)\begin{pmatrix} 1 \\ 0 \end{pmatrix},\tag{3}$$

where $\alpha_1$ and $C_1$ are the orientation of the slow optic axis and the phase retardation inside the first layer, respectively. We can generalize the subscript by using $n$ for the $n$-th layer. Subsequently, these two parameters depend on terahertz E-field vector at $Z = \frac{n-1}{N}l$. We assume that the polarization state of the probe pulse in front of the EO crystal is linearly polarized along the $X$-direction. After passing all $N$ layers, the polarization state of the probe pulse can be written as

$$\begin{pmatrix} E_X \\ E_Y \end{pmatrix} = \prod_n^N \left[ R(-\alpha_n)\begin{pmatrix} \exp\left(-\frac{iC_n}{2}\right) & 0 \\ 0 & \exp\left(\frac{iC_n}{2}\right) \end{pmatrix} R(\alpha_n) \right]\begin{pmatrix} 1 \\ 0 \end{pmatrix}$$
$$\cong \begin{pmatrix} 1 - \sum iC_n \cos 2\alpha_n \\ -\sum iC_n \sin 2\alpha_n \end{pmatrix}.\tag{4}$$

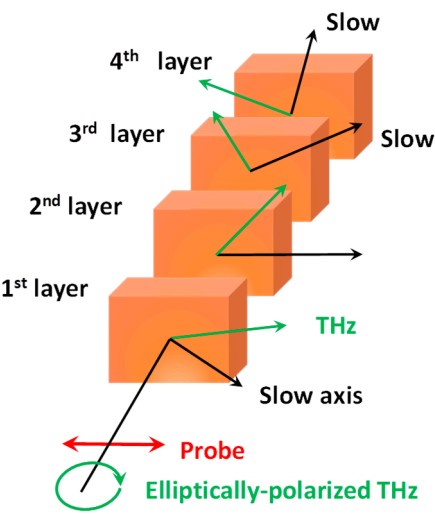

**Figure 1.** Schematic illustration of the multilayer description to interpret the detected electro-optic (EO) signal for elliptically-polarized terahertz waves, including the effect of the velocity mismatch between terahertz wave and probe pulse.

For the last equality, we assumed that the retardation $C_n$ in each layer is much smaller than unity ($C_n \ll 1$). Equation (4) shows that the total birefringence can be expressed as a superposition of the terahertz E-field-induced birefringence in each layer. The EO signal $S(\tau)$ is proportional to the $Y$-component in Equation (4) ($\sum C_n \sin 2\alpha_n$). In the limit of $N \to \infty$, the EO signal $S(\tau)$ becomes

$$S(\tau) \propto \int_0^l C(Z, \tau) \sin 2\alpha(Z, \tau) dZ,\tag{5}$$

where $\alpha(Z, \tau)$ and $C(Z, \tau)$ are the orientation of the slow optic axis and retardation at position $Z$ for a probe pulse with a time delay of $\tau$. Because these parameters depend on both the terahertz E-field vector and orientation of the EO crystal via the terahertz E-field-induced birefringence, we can obtain information on the terahertz E-field vector by employing the proper angle of the EO crystal. For example, when a <110>-oriented zinc-blende crystal (point group $\bar{4}3$ m) is utilized, Equation (5) can be rewritten as [98]

$$S(\tau) \propto r_{41} \int_0^l E_1(Z, \tau)[\cos(\varphi + \gamma(Z, \tau)) + 3\cos(3\varphi - \gamma(Z, \tau))]dZ,\tag{6}$$

where $E_1(Z,\tau)$ and $\gamma(Z,\tau)$ are the amplitude and orientation of the terahertz E-field vector, respectively, and $\varphi$ is the angle of the $[\overline{1}10]$ crystal direction of the zinc-blende structure with respect to the *X*-axis. Equation (6) describes the EO signal in the multilayer model that is obtained when an elliptically-polarized terahertz E-field, including velocity mismatch is measured. The EO signal $S(\tau)$ can be expressed as an integral of the E-field vector that is sensed by the probe pulse over *Z*.

In order to compare Equation (6) with the previous result [81] (see Equation (2)), we introduce the double integration over position *Z* and time *t* by using the relation $I_{\mathrm{OPT}}(Z, t-\tau) \propto \delta(t - \tau - Z/v_{\mathrm{g}})$:

$$S(\tau) \propto r_{41} \int_0^l dZ \int_{-\infty}^{+\infty} dt\, I_{\mathrm{OPT}}(Z, t-\tau) E_1(Z,t) [\cos(\varphi + \gamma(Z,t)) + 3\cos(3\varphi - \gamma(Z,t))]. \qquad (7)$$

Here, we stress that this description is only applicable to delta-function-like probe pulses. Hence, one cannot utilize any other function for $I_{\mathrm{OPT}}(Z, t-\tau)$ in Equation (7). Equation (7) is the final result of the multilayer model to interpret the measured EO signal for a general terahertz wave, including velocity mismatch. This result contains the situations considered in previous literatures as special cases. For example, in the perfect velocity-matching condition that has been treated in Refs. [36,68], $E_{\mathrm{THz}}$ and $\gamma$ have constant values throughout the EO crystal. In this situation, Equation (7) can be written as

$$S(\tau) \propto r_{41} \cdot l \cdot E_1(\tau) [\cos(\varphi + \gamma(\tau)) + 3\cos(3\varphi - \gamma(\tau))], \qquad (8)$$

which is the same equation as shown in Refs. [36,68]. If the terahertz E-field is linearly-polarized as considered in Ref. [81], the orientation of the terahertz E-field vector ($\gamma$) can be treated as a constant. Under this condition, Equation (7) becomes $S(\tau) \propto r_{41} \int_0^l dZ \int_{-\infty}^{+\infty} dt\, I_{\mathrm{OPT}}(Z, t-\tau) E_1(Z,t)$, which is Equation (2) derived by Bakker et al. [81]. These comparisons clarify the validity of our multilayer model.

## 2.2. Frequency-Domain Description

In the previous section, we introduced the multilayer model based on the Pockels effect description to intuitively describe the EO signal that is obtained for an elliptically-polarized terahertz pulse, including the effect of a velocity mismatch. One of the weak points of this model is that we have to assume a delta-function-like ultra-short pulse in order to omit the complicated effect of the simultaneous influence of different terahertz E-field vector directions within the probe pulse width. However, when we want to take into account for this effect, we have to introduce the frequency-domain description. In this section, we explain the frequency-domain description to interpret the EO signal that is obtained when an elliptically-polarized terahertz E-field impinges on the EO crystal in the presence of a velocity mismatch.

Figure 2 shows the difference between the Pockels effect (described in the previous section) and frequency-domain descriptions. The latter was initially proposed by Gallot and Grischkowsky [83] for interpretation of the results in the case of linearly-polarized terahertz pulses. In terms of the Pockels effect (Figure 2a), the polarization direction of the linearly-polarized probe pulse rotates inside the EO crystal because of the terahertz E-field-induced birefringence. On the other hand, in the frequency-domain description (Figure 2b), the polarization of the probe pulse (whose Fourier component is denoted by $E_{2X}(\omega_2)$) stays unchanged inside the EO crystal. Instead of the terahertz E-field-induced birefringence, we consider the additional frequency mixing fields, i.e., the sum frequency generation (SFG) field and difference frequency generation (DFG) field, caused by the nonlinear polarization field that is induced by the terahertz and probe pulses. These fields are added to the probe pulse and thus result in a change of the measured polarization direction. We denote the *X*- and *Y*-components of the summed SFG and DFG fields by $E_{3X}(\omega_3)$ and $E_{3Y}(\omega_3)$, respectively. Subsequently, the relation $\omega_3 = \omega_2 \pm \Omega$ is satisfied where $\Omega$ is the angular frequency of terahertz field. We assume that the frequency bandwidth of the probe pulse is larger than $\Omega$ and therefore the frequency components of the probe, SFG, and DFG pulses overlap with each other, as shown in the inset of Figure 2. The Jones vector of the total E-field, including the probe, SFG, and DFG fields at angular frequency $\omega_3$, can be described by

$$\begin{pmatrix} E_{2X}(\omega_3) + E_{3X}(\omega_3) \\ E_{3Y}(\omega_3) \end{pmatrix}. \tag{9}$$

Equation (9) corresponds to the Jones vector $\begin{pmatrix} 1 - iC\cos 2\alpha \\ -iC\sin 2\alpha \end{pmatrix}$ in the previous description based on the Pockels effect. In the conventional EO sampling setup, the EO signal $S$ can be described in the frequency-domain description as [93]

$$S \propto \int_{-\infty}^{+\infty} i \cdot \text{sign}(\omega_3) \cdot E_{2X}^*(\omega_3) E_{3Y}(\omega_3)\, d\omega_3, \tag{10}$$

where $\text{sign}(\omega_3)$ is $+1(-1)$ if $\omega_3 > 0$ ($\omega_3 < 0$) and $E_{2X}^*(\omega_3)$ is the complex conjugate of $E_{2X}(\omega_3)$. A comparison between Equations (4) and (10) reveals that, in both descriptions, the EO signal $S$ is proportional to the $Y$-component of the optical fields after passing through the EO crystal. This is a characteristic of the conventional balanced detection scheme of the EO sampling [81], independent of the type of description employed.

## (a) Pockels description  (b) Frequency-domain description

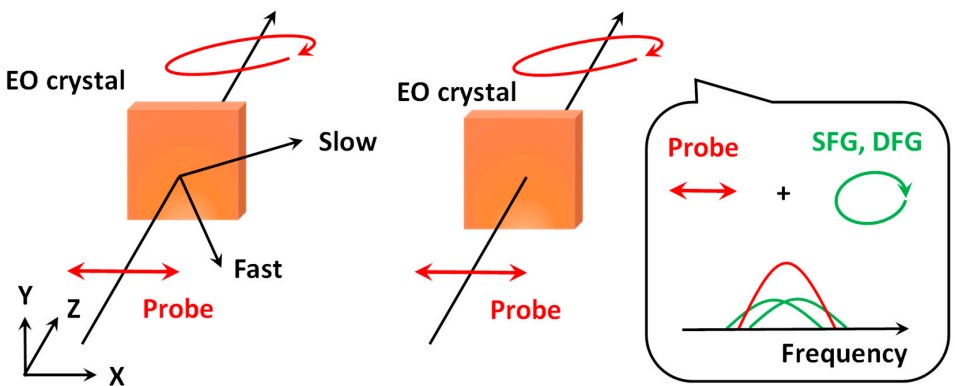

**Figure 2.** The difference between (**a**) the Pockels effect and (**b**) frequency-domain descriptions. The inset shows that the spectra of sum and difference frequency fields (green) have a large overlap with the probe spectrum (red), because the terahertz frequency is much smaller than the frequency bandwidth of the probe pulse. The polarization state of each frequency component is generally elliptically-polarized and can be expressed as a superposition of probe, sum, and difference frequency fields. EO: electro-optic, SFG: sum frequency generation, DFG: difference frequency generation.

Next, we show that the frequency-domain description naturally yields the PS-EO signal without the need of considering a delta-function-like pulse shape of the probe pulse. In the multilayer description in the previous section, we had to cope with the problem that the E-field vector of a probe pulse with a finite width temporally should change within the pulse width due to the position-dependent direction of the slow optic axis, as shown in Figure 1. Because we cannot treat such a complex situation, we assumed that the probe pulse has delta-function-like pulse shape. On the other hand, in the frequency-domain description, we assume that the polarization state of the probe pulse itself ($E_{2X}(\omega_3)$) is not affected by the propagation through the EO crystal. Therefore, it stays linearly-polarized along the $X$-direction even if it has a finite pulse width. However, the polarization state of the SFG and DFG fields is elliptical, as shown in the inset of Figure 2b, which induces the actual PS-EO signal. Indeed, instead of regarding the PS-EO signal as polarization rotation of the probe pulse, we regard it as the consequence of the additional SFG and DFG fields that appear in the

frequency-domain description. Because the frequency-domain description does not depend on the pulse shape of the probe pulse, we can develop a general formulation of the PS-EO signal.

As outlined above, the polarization information of the terahertz wave is included in the SFG and DFG fields. The SFG and DFG fields have their origin in the nonlinear polarization $P_3(\omega_3 = \omega_2 + \Omega)$, which is generally described as $P_3(\omega_3) = \chi^{(2)}(\omega_3 = \omega_2 + \Omega) : E_1(\Omega)E_2(\omega_2)$, where $\chi^{(2)}(\omega_3)$ is the second-order nonlinear susceptibility tensor, and $E_1(\Omega)$ and $E_2(\omega_2)$ are single frequency components of the terahertz and probe pulses, respectively (the notation employed here is based on Shen's textbook [99]). Because the probe pulse is linearly polarized, only the $X$-component of $E_2(\omega_2)$, $E_{2X}(\omega_2)$, is nonzero. Nonvanishing elements as well as the equal elements of the tensor $\chi^{(2)}$ depend on the symmetry class of the chosen EO crystal [99–101]. Therefore, from the symmetry class of a certain EO crystal and the polarization state of $E_1(\Omega)$, we can derive the direction of the vector $P_3(\omega_3)$. By calculating the oscillating profile of $P_3(\omega_3)$, we can derive the polarization states of the SFG and DFG fields whose sum is denoted by $E_3(\omega_3)$. Note that in the Pockels effect description, it is considered that the EO signal is induced by the instant E-field of the terahertz pulse where relations $\Omega = 0$ and $\omega_3 = \omega_2$ are always satisfied. In contrast, the frequency-domain description can treat the general case where the terahertz frequency $\Omega$ has finite values, which is essential to precisely retrieve the frequency-dependent polarization of the terahertz pulse.

Here, for example, we utilize a <110>-oriented zinc-blende EO crystal with the $\overline{4}3m$ point group symmetry (the complete mathematical description is found in our previous paper [93]). We define $\varphi$ as the angle between the $[\overline{1}10]$ direction of the crystal and the $X$-axis. In this case, the $Y$-component of the second-order nonlinear polarization $P_{3Y}(\omega_3)$ is proportional to $d(\omega_3; \Omega, \omega_3 - \Omega)E_{1X}(\Omega)E_{2X}(\omega_3 - \Omega)$ at $\varphi = 0°$ and proportional to $d(\omega_3; \Omega, \omega_3 - \Omega)E_{1Y}(\Omega)E_{2X}(\omega_3 - \Omega)$ at $\varphi = 90°$, where $d$ represents the nonlinear coefficient for zinc-blende crystals with $d \equiv d_{14} = d_{25} = d_{36}$ [100,101]. Because total $Y$-component of the SFG and DFG fields, $E_{3Y}$, is derived from $P_{3Y}$, we can obtain $E_{3Y}$ by solving the wave equation [99]. Finally, we obtain the following expressions of $E_{3Y}$ for the two different crystal orientations [93]:

$$E_{3Y}(\omega_3) \propto \int_{-\infty}^{+\infty} d(\omega_3; \Omega, \omega_3 - \Omega) \frac{\exp(i\Delta k(\Omega, \omega_3 - \Omega, \omega_3))}{i\Delta k(\Omega, \omega_3 - \Omega, \omega_3)} E_{2X}(\omega_3 - \Omega)E_{1X}(\Omega)d\Omega \ (\varphi = 0°), \quad (11)$$

and

$$E_{3Y}(\omega_3) \propto \int_{-\infty}^{+\infty} d(\omega_3; \Omega, \omega_3 - \Omega) \frac{\exp(i\Delta k(\Omega, \omega_3 - \Omega, \omega_3))}{i\Delta k(\Omega, \omega_3 - \Omega, \omega_3)} E_{2X}(\omega_3 - \Omega)E_{1Y}(\Omega)d\Omega \ (\varphi = 90°), \quad (12)$$

where $\Delta k(\Omega, \omega_3 - \Omega, \omega_3) \equiv k_1(\Omega) + k_2(\omega_3 - \Omega) - k_3(\omega_3)$, and $k_i$ ($i$ assumes the values 1, 2, and 3) are complex wavenumbers. The subscripts 1, 2, and 3 represent the terahertz pulse, probe pulse, and the combined SFG and DFG pulse, respectively. As shown in Equations (11) and (12), $E_{3Y}(\omega_3)$ includes the information of the terahertz E-field in either $X$- or $Y$-direction ($E_{1X}(\Omega)$ or $E_{1Y}(\Omega)$) depending on the EO crystal angle $\varphi$. Because the EO signal is a function of $E_{3Y}(\omega_3)$ as shown in Equation (9), we can derive the polarization information of the terahertz wave ($E_{1X}(\Omega)$ and $E_{1Y}(\Omega)$) from the EO signal by simply rotating the EO crystal. The EO signals at the delay time $\tau$ between terahertz and probe pulses can be derived from Equations (10)–(12) and become [93]

$$S_X(\tau) \equiv S(\tau)(\varphi = 0°) \propto \int_{-\infty}^{+\infty} f(\Omega)E_{1X}(\Omega)\exp(-i\Omega\tau)d\Omega, \quad (13)$$

and

$$S_Y(\tau) \equiv S(\tau)(\varphi = 90°) \propto \int_{-\infty}^{+\infty} f(\Omega)E_{1Y}(\Omega)\exp(-i\Omega\tau)d\Omega . \quad (14)$$

Here, $f(\Omega)$ is the frequency-filtering function, which incorporates the effects of probe pulse width, phase mismatch, absorption, and the frequency-dependent nonlinear optical coefficient [83,93]. The two EO signals $S_X(\tau)$ and $S_Y(\tau)$ can be considered as the time-domain profiles of the $X$- and

*Y*-components of the E-field of the elliptically-polarized terahertz pulse modulated by the frequency filtering function $f(\Omega)$ for each frequency $\Omega$ in the integral. (Please note that when $f(\Omega) = 1$, Equation (13) (Equation (14)) exactly represents the time domain profile of the *X*- (*Y*-) component of the E-field of the elliptically-polarized terahertz pulse). The striking advantage of the frequency-domain description is that the effects of the finite pulse width, phase mismatch, absorption losses, and the frequency-depend nonlinear coefficient are naturally included in the expressions for the EO signal through $f(\Omega)$. This is in stark contrast to the multilayer model, where it is difficult to include all of these effects in a simple picture.

Equations (13) and (14) represent the results for <110>-oriented zinc-blende crystals. It is noted that these equations can also be applied to other EO crystals with different orientations and/or crystal symmetries, e.g., <111>-oriented zinc-blende crystals, and *c*-cut GaSe and LiNbO$_3$ crystals with normal incidence [94]. In these cases, the angle $\varphi$ is defined either as the orientation of the $[\overline{2}11]$ crystal direction for the <111>-oriented zinc-blende crystal, or as the angle of the *x*-axis for *c*-cut GaSe and LiNbO$_3$ crystals with respect to the polarization direction of the probe pulse (*X*-axis). Subsequently, the same results as shown in Equations (13) and (14) are derived.

### 2.3. Precise Polarization Spectroscopy with Aid of PS-EO Sampling

In the previous section, we showed that the PS-EO signal is not exactly the original time-domain profile of the *X*- and *Y*-component of the E-field transients of the elliptically-polarized terahertz pulse entering the EO crystal, but more or less distorted because of the frequency filtering function $f(\Omega)$, as shown in Equations (13) and (14). Nevertheless, the polarization information of the original elliptically-polarized terahertz pulse can be easily retrieved from the PS-EO signal, irrespective of the degree of the distortion [95]. In this section, we explain the details concerning this important aspect of the frequency-domain description for the PS-EO sampling.

The polarization state of an elliptically-polarized terahertz E-field with angular frequency $\Omega$ can be characterized by the ellipticity angle $\theta(\Omega)$ and the angle of rotation $\Psi(\Omega)$. By introducing a new complex parameter $\chi(\Omega) \equiv E_{1Y}(\Omega)/E_{1X}(\Omega)$, the $\theta(\Omega)$ and $\Psi(\Omega)$ can be derived via the following relations [101]:

$$\tan 2\Psi(\Omega) = \frac{2\mathrm{Re}[\chi(\Omega)]}{1 - |\chi(\Omega)|^2}, \tag{15}$$

and

$$\sin 2\theta(\Omega) = \frac{2\mathrm{Im}[\chi(\Omega)]}{1 + |\chi(\Omega)|^2}. \tag{16}$$

Thus, the experimental evaluation of $\chi(\Omega)$ is sufficient to determine the polarization state of the elliptically-polarized terahertz pulse. The parameter $\chi(\Omega)$ can be easily derived from the PS-EO signals $S_X(\tau)$ and $S_Y(\tau)$ given in Equations (13) and (14). By performing the Fourier transform of $S_X(\tau)$ and $S_Y(\tau)$, we obtain $S_X(\Omega)$ and $S_Y(\Omega)$, respectively. From Equations (13) and (14), it is found that $S_X(\Omega) \propto f(\Omega)E_{1X}(\Omega)$ and $S_Y(\Omega) \propto f(\Omega)E_{1Y}(\Omega)$. Therefore, the common parameter $f(\Omega)$ cancels when we divide $S_Y(\Omega)$ by $S_X(\Omega)$. Hence, we can evaluate $\chi(\Omega)$, also via

$$\chi(\Omega) = \frac{S_Y(\Omega)}{S_X(\Omega)}. \tag{17}$$

Equation (17) is the key of the terahertz polarization spectroscopy based on PS-EO sampling, because it shows that the parameter $\chi(\Omega)$ is independent of $f(\Omega)$, and thus can be derived from the simple division of the Fourier transformed PS-EO signals. As mentioned above, the temporal profiles of the PS-EO signals differ from the original time-domain profile of the elliptically-polarized terahertz pulse due to $f(\Omega)$. Because $f(\Omega)$ depends on many complicated effects such as the finite probe-pulse width, degree of phase mismatching, absorption coefficient, and frequency-dependent nonlinear optical coefficient, the interpretation of the time-domain waveform is rather difficult. On the

other hand, the polarization state of each frequency component is independent of $f(\Omega)$, and thus it can be simply derived from Equation (17). Section 3.1 provides the experimental demonstration of the polarization spectroscopy based on this finding.

*2.4. Retrival of Elliptically-Polarized Terahertz Time-Domain E-Field Waveforms*

In some applications, such as terahertz nonlinear spectroscopy [102,103], not only the polarization state but also the exact temporal profile of the E-field vector of the terahertz pulse is required to fully understand the optical responses. To analyze the E-field vector time-domain waveform, we should consider the effect of $f(\Omega)$ on the measured data. In this section, we explain how the frequency-domain description can be used to retrieve elliptically-polarized terahertz time-domain waveforms from the Fourier transforms of the measured PS-EO sampling data, $S_X(\Omega)$ and $S_Y(\Omega)$.

The temporal profile of the elliptically-polarized terahertz pulse in front of the EO crystal, $\mathbf{E}_1(\tau)$, which we want to retrieve, is described by [95]

$$\mathbf{E}_1(\tau) = \int_{-\infty}^{+\infty} \left(\frac{2}{1+\hat{N}(\Omega)}\right)^{-1} \left[E_{1X}(\Omega) \cdot \hat{X} + E_{1Y}(\Omega) \cdot \hat{Y}\right] \exp(-i\Omega\tau)d\Omega, \tag{18}$$

where $\hat{N}(\Omega)$ is the complex refractive index of the EO crystal and $\hat{X}$ ($\hat{Y}$) is a unit vector along the $X$- ($Y$-) direction. The term $\left(\frac{2}{1+\hat{N}(\Omega)}\right)^{-1}$ corresponds to the reflection of the terahertz pulse at the front surface of the EO crystal. By substituting the relations $S_X(\Omega) \propto f(\Omega)E_{1X}(\Omega)$ and $S_Y(\Omega) \propto f(\Omega)E_{1Y}(\Omega)$ into Equation (18), we obtain

$$\mathbf{E}_1(\tau) \propto \int_{-\infty}^{+\infty} \left(\frac{2}{1+\hat{N}(\Omega)}\right)^{-1} f^{-1}(\Omega)\left[S_X(\Omega) \cdot \hat{X} + S_Y(\Omega) \cdot \hat{Y}\right] \exp(-i\Omega\tau)d\Omega. \tag{19}$$

Although Equation (19) is an apparently simple equation, the retrieval process of $\mathbf{E}_1(\tau)$ with Equation (19) is actually very complicated because of the complexity of $f(\Omega)$. In order to perform the calculation, some researchers have approximated $f(\Omega)$ using well-defined optical parameters, such as dispersion, absorption, the frequency-dependent nonlinear optical coefficient [83,85], and the pulse shape of the probe pulse [88]. We used these previous results and developed a new approximation for $f(\Omega)$ that takes into account the effects of the phase mismatch, finite pulse width of the probe pulse, and absorption of the terahertz wave in the EO crystal, as shown in Equation (20) below [95].

$$f(\Omega) \approx \frac{1}{\sqrt{a^2(\Omega)+\beta_1^2(\Omega)}} \left|\chi^{(2)}(\omega_0, \Omega, \omega_0 - \Omega)\right| \exp\left[-\frac{\Omega^2\tau_p^2}{4}\right] \exp\left[-\{\beta_1(\Omega) - ia(\Omega)\}\frac{l}{2}\right]$$
$$\times \sqrt{\cos^2\left(\frac{a(\Omega)l}{2}\right)\sinh^2\left(\frac{\beta_1(\Omega)l}{2}\right) + \sin^2\left(\frac{a(\Omega)l}{2}\right)\cosh^2\left(\frac{\beta_1(\Omega)l}{2}\right)}. \tag{20}$$

Here, we employed the following definitions: $a(\Omega)$ describes the phase mismatch between the terahertz wave (with real wavenumber $k_1^R(\Omega)$) and the probe pulse (with group velocity $v_g$) and it is defined by $a(\Omega) \equiv k_1^R(\Omega) - \Omega/v_g$. Further, $\beta_1(\Omega)$ is the frequency-dependent absorption coefficient, $\left|\chi^{(2)}\right|$ is the second-order nonlinear optical susceptibility, and $\tau_p$ is the pulse width of the probe pulse. These parameters can be experimentally evaluated in separate experiments. Here, we assumed that the loss of the probe pulse inside the EO crystal is negligible. Note that the case accounting for the loss of the probe pulse is given in Appendix A of Ref. [95].

Next, we discuss the relations between Equation (20) and some previous works. When $\beta_1(\Omega)$ is zero, the square-root terms of Equation (20) is proportional to $\sin c(a(\Omega)l/2)$. The same sinc-function term is also found in the frequency mixing signal that accounts for the phase mismatch without absorption losses [100]. The term $\exp\left[-\frac{\Omega^2\tau_p^2}{4}\right]$ appears in Ref. [84,87] as the effect of the finite pulse width. When considering these relations, we conclude that all of the information regarding

the phase mismatch, absorption, and finite pulse width are included in Equation (20). This means that we can accurately retrieve terahertz time-domain waveform, because "all the information" are contained in Equation (20). Section 3.2 demonstrates the retrieval of the terahertz time-domain E-field vector waveform.

A brief summary of the above theoretical formulation for the retrieval of the terahertz time-domain E-field waveform is given in the following. Equation (19) describes the relationship between the original elliptically-polarized terahertz time-domain E-field vector waveform before impinging on the EO crystal, $\mathbf{E}_1(\tau)$, and the measured intensity difference signal obtained by the PS-EO sampling method (the Fourier transform of $S_X(\Omega)$ and $S_Y(\Omega)$). This equation is derived in the framework of the frequency-domain description and it accounts for the effects of the finite pulse width of the probe pulse and the optical properties of the EO crystal, such as dispersion, frequency-dependent absorption, and nonlinear optical coefficients. Therefore, we can precisely retrieve the original terahertz time-domain E-field vector waveform if the abovementioned parameters have been determined experimentally. On the other hand, although the multilayer model that is described in Section 2.1 provides an intuitive picture of the measured E-field time traces using the PS-EO sampling method (see Equation (6)), it is very difficult to mathematically formulate the procedure required for retrieving the original time-domain terahertz E-field vector waveform from the experimental data. In particular, it is too difficult to include the effect of the finite pulse width of the probe pulse in this description, because we would have to consider that the terahertz E-field-induced birefringence changes within the probe pulse. We consider that the frequency-domain description is more suited for terahertz polarization spectroscopy, providing a full understanding of the experimental data measured by the PS-EO sampling method.

## 3. Experiment

The experimental demonstration of our proposed implementation of the terahertz polarization spectroscopy is presented in Section 3.1, and the retrieval of the elliptically-polarized terahertz time-domain waveforms is demonstrated in Section 3.2. Both experimental results strongly support the usefulness of the theoretical formulation based on the frequency-domain description given in the previous section.

First, we briefly explain our experimental setup. A near infrared pulse with a center wavelength of 800 nm is emitted from a Titanium:sapphire laser with a repetition rate of 80 MHz and a pulse duration of about 90 fs. The output pulse is divided into pump and probe pulses for the generation and detection of the terahertz pulse, respectively. For the measurement of the temporal evolution of the terahertz E-field, we change the relative time interval between the terahertz and probe pulses by moving a mechanical delay stage that is placed in the pump beam path. The terahertz pulses generated from a ZnTe crystal include frequency components approximately ranging from 0.5 to 2.5 THz. To minimize the effect of water vapor absorption on the terahertz pulses, dry air is used in the part of the setup where the terahertz pulse propagates. To produce a well-defined elliptically-polarized terahertz pulse, we insert a wire-grid polarizer and a monochromatic quartz QWP (Tydex, operation wavelength 496 μm) in the terahertz beam path. The E-field time trace is detected by the commonly used balanced detection scheme [96]. More details of our experimental setup are provided in Refs. [93–95].

### 3.1. Polarization Spectroscopy

In this section, we use polarization spectroscopy measurements to verify the validity of our theoretical formulation that is described in Section 2.3. We conclude that since the frequency-filtering function $f(\Omega)$ is canceled by the division of the orthogonal signal components, the polarization state of each frequency component can be obtained, irrespective of the chosen EO crystal. To confirm this conclusion, we prepare several EO crystals, record the EO signals with the same elliptically-polarized terahertz pulse and experimental conditions, and compare the obtained polarization parameters, i.e., $\theta(\Omega)$ and $\Psi(\Omega)$.

We prepare six EO crystals: two <110>-oriented ZnTe crystals with thicknesses of 1 and 2 mm, a <110>-oriented GaP crystal with a thickness of 0.4 mm [93], a <111>-oriented ZnTe crystal with a thickness of 1 mm, a 0.1-mm-thick c-cut GaSe crystal, and a 0.5-mm-thick c-cut LiNbO$_3$ crystal [94]. For each of these six EO crystals, the auxiliary angle $\theta(\Omega)$ of the ellipticity and the angle of rotation $\Psi(\Omega)$ of the elliptically-polarized terahertz pulse are plotted as a function of frequency in either Figure 3a,b or Figure 4a,b. In each of these figures, the obtained data points almost coincide with each other. The obtained $\theta(\Omega)$ spectra that are shown in Figures 3a and 4a are well explained by the frequency-dependent polarization response of the monochromatic QWP. Since the used QWP is designed for 496 μm (~0.6 THz), the polarization state at 0.6 THz should be circularly-polarized, i.e., $\theta \sim \pi/4$. In addition, the QWP is expected to act as half, three-forth, and full waveplate at 1.2, 1.8, and 2.4 THz, respectively. Thus, the polarization state should become linearly-polarized, circularly-polarized, and again linearly-polarized at ~1.2, ~1.8, and 2.4 THz, respectively. The experimental results that are shown in Figures 3a and 4a are in good agreement with this predicted frequency dependence of the ellipticity angle. In addition, the frequency dependence of $\Psi$ is also well explained by the frequency-dependent polarization response of the QWP as well as that of $\theta$. These consistent results strongly support our conclusion that the polarization state of the terahertz wave can be obtained precisely with any of the chosen EO crystals. The difference in $\theta$ and $\Psi$ between Figures 3 and 4 is due to different experimental conditions between Refs. [93,94]. A more detailed polarization analysis using Stokes parameters and the Poincaré sphere representation is provided in our previous work [93].

Finally, we would like to comment on the measurable frequency range of the PS-EO sampling method. Recently, many researchers have demonstrated the measurements of mid- and near-infrared E-field waveforms by the EO sampling method with ultrashort pulses and a variety of EO crystals [15–24]. Because our formalism is independent of the choice of the EO crystal, such ultrashort pulses and the appropriate selection of EO crystals enables the application of the PS-EO sampling method in an ultra-broadband frequency range.

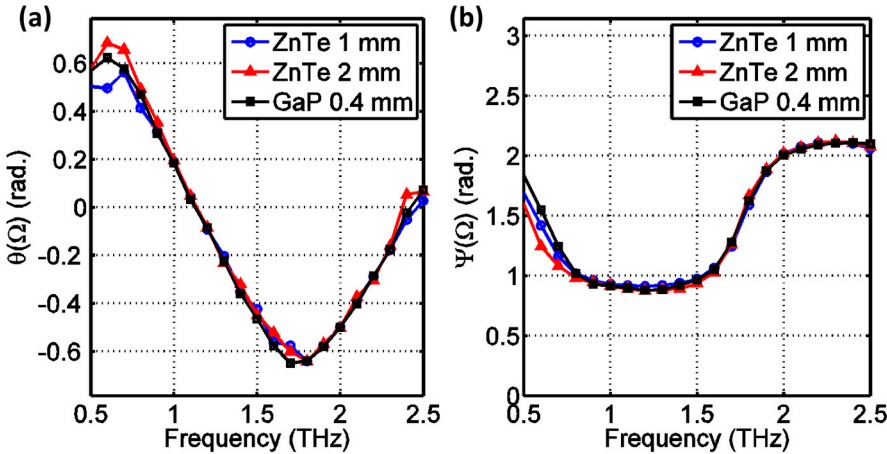

**Figure 3.** (**a**) The auxiliary angle $\theta(\Omega)$ of the ellipticity and (**b**) the angle of rotation $\Psi(\Omega)$ of the elliptically-polarized terahertz pulse measured by using three different ⟨110⟩-oriented zinc-blende crystals [1-mm-thick ZnTe crystal (circles), 2-mm-thick ZnTe crystal (triangles), and a 0.4-mm-thick GaP crystal (squares)]. Lines are guides to the eye. Reprinted with permission from Ref. [93], OSA publishing (J. Opt. Soc. Am. B **31**, 3170-3180 (2014)).

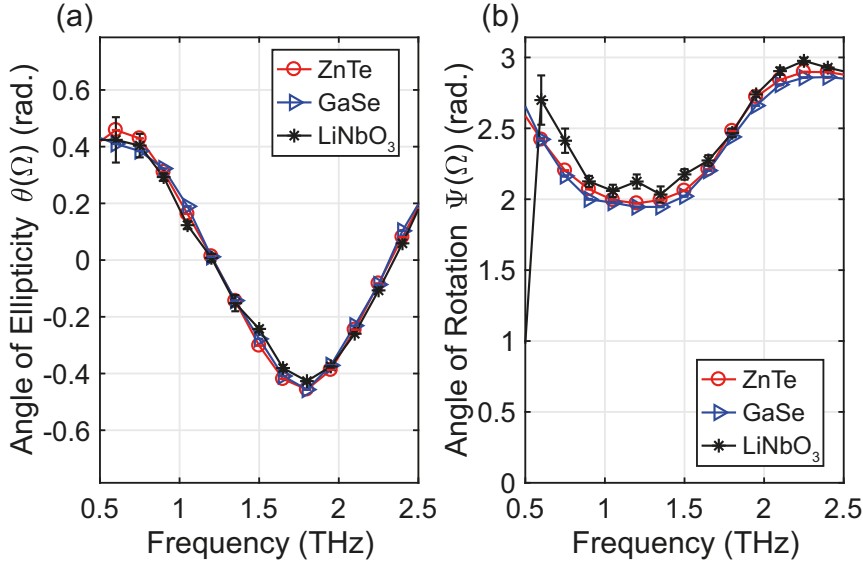

**Figure 4.** (**a**) The auxiliary angle $\theta(\Omega)$ of the ellipticity and (**b**) the angle of rotation $\Psi(\Omega)$ of the elliptically-polarized terahertz pulse measured by using three EO crystals with different crystal symmetries [1-mm-thick $\langle 111 \rangle$-oriented ZnTe (open circles), 0.1-mm-thick c-cut GaSe (open triangles), and a 0.5-mm-thick c-cut LiNbO$_3$ (asterisks)]. Lines are guides to the eye. The length of the error bar at each point is twice the standard deviation of the mean estimated from ten repeated experiments. Reproduced from [Appl. Phys. Lett., Vol. **108**, Issue 1, pp. 011105 (2016)], with permission of AIP Publishing.

*3.2. Retrieval of the Elliptically-Polarized Terahertz Time-Domain Waveform*

In this section, we show experimental results regarding the retrieval of the elliptically-polarized terahertz time-domain waveform to verify our theoretical formulation that is described in Section 2.4. Here, we measure the same elliptically-polarized terahertz pulse by using two different <110>-oriented EO crystals, a 1 mm-thick ZnTe and a 0.4 mm-thick GaP crystal, and retrieve the elliptically-polarized terahertz time-domain E-field waveform according to the method that is described by Equation (19).

First, to calculate $f(\Omega)$ in Equation (20), we estimate the essential parameters $a(\Omega)$, $\beta_1(\Omega)$, $\left| \chi^{(2)} \right|$, and $\tau_p$. For evaluating $a(\Omega)$ and $\beta_1(\Omega)$, we measure the complex refractive indices of the two EO crystals in the terahertz frequency range and the group velocities of the probe pulse inside the EO crystals. The obtained values of the complex refractive indices and the group velocities for the two EO crystals are similar to those reported in previous works [14,104,105]. Since $\left| \chi^{(2)} \right|$ is proportional to $r_{41}$, we used the values $r_{41} = 4 \, \mathrm{pm/V}$ for ZnTe and $r_{41} = 1 \, \mathrm{pm/V}$ for GaP [14]. In addition, we use the pulse width of the light source as $\tau_p$ (approximately 90 fs).

Next, we measure the EO signals $S_X(\tau)$ and $S_Y(\tau)$ by using the two EO crystals. Figure 5a,b show the E-field time traces that were obtained with the ZnTe crystal (blue data) and the GaP crystal (red data). The magnitudes of $S_X(\tau)$ and $S_Y(\tau)$ for the GaP crystal are magnified by a factor of 7 for clarity. Since we measure the same elliptically-polarized terahertz pulse, we can infer that the difference in the measured time-domain waveforms is due to the difference in the $f(\Omega)$ of the two EO crystals. The difference in the magnitude is attributed to the difference in $|f(\Omega)|$. In the ZnTe and GaP crystals, the difference in $|f(\Omega)|$ is mainly governed by the difference in $\left| \chi^{(2)} \right| \propto r_{41}$, i.e., $r_{41} = 4 \, \mathrm{pm/V}$ in ZnTe and $r_{41} = 1 \, \mathrm{pm/V}$ in GaP. On the other hand, the time-domain signal measured by the GaP crystal arrives earlier than that measured by the ZnTe crystal. This fact can be explained by the difference in $\mathrm{Arg}[f(\Omega)] = \mathrm{Arg}[\exp(-ia(\Omega)l/2)]$, which is determined by the crystal thickness and the refractive index.

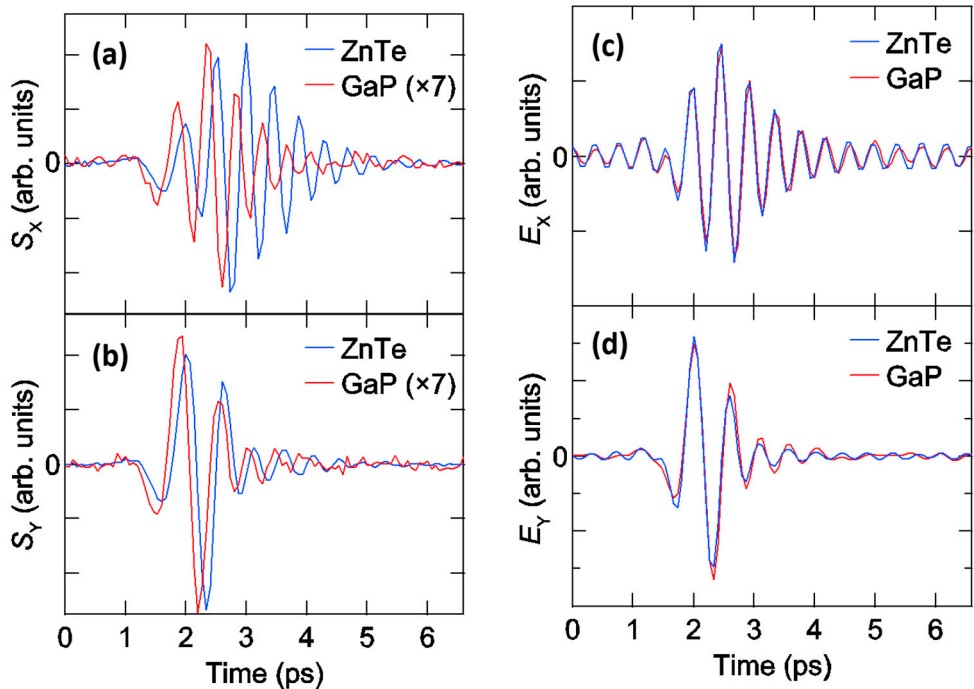

**Figure 5.** (**a**) $S_X(\tau)$ and (**b**) $S_Y(\tau)$ measured by the 1-mm-thick $\langle 110 \rangle$-oriented ZnTe (blue curves) and the 0.4-mm-thick $\langle 110 \rangle$-oriented GaP (red curves) crystals. The magnitude of the signals measured by the GaP crystal is multiplied by 7 for easy comparison. (**c**) The original $E_X(\tau)$ and (**d**) $E_Y(\tau)$ retrieved from the EO signals measured by the 1-mm-thick $\langle 110 \rangle$-oriented ZnTe (blue curves) and the 0.4-mm-thick $\langle 110 \rangle$-oriented GaP (red curves) crystals. Reprinted with permission from Ref. [95], OSA publishing (J. Opt. Soc. Am. B **34**, 1946-1956 (2017)).

Figure 5c,d show the retrieved $E_X(\tau)$ and $E_Y(\tau)$, calculated from the $S_X(\tau)$ and $S_Y(\tau)$ shown in Figure 5a,b according to Equation (19). We emphasize that no additional signal processing has been performed to obtain this result. Nevertheless, the two retrieved time-domain waveforms show a very good agreement! For completeness, we briefly comment on the small oscillation before and after the terahertz pulse shown in Figure 5c. We concluded that this oscillation is an artifact from the signal processing due to the limited generation and detection bandwidth. We stress that the very good agreement between the retrieved time-domain waveforms (which were measured by two different EO crystals) strongly indicates the validity of our retrieval method based on the frequency-domain description.

## 4. Conclusions

In this invited review, we focus on our recent progress in the PS-EO sampling when an elliptically-polarized terahertz pulse impinges on the EO crystal. We introduce two descriptions to interpret the EO signal: the multilayer model and the frequency-domain description. The multilayer model provides a straightforward understanding of the measured PS-EO signal, with the drawback that it is difficult to retrieve the original time-domain waveform. The frequency-domain description is theoretically more rigorous, and the effects of the finite pulse width of the probe pulse, the phase mismatch, absorption, and the frequency dependence of the nonlinear optical susceptibility of the EO crystal are naturally taken into account.

The two main conclusions of this review provided below are proven by theory and experiment. Firstly, the polarization state of each frequency component can be accurately measured, irrespective of the choice of the EO crystal, although a different crystal implies a different frequency filtering function. Secondly, the distortion effects that appear in the measured E-field time traces are characterized by

the frequency filtering function, and we propose a proper retrieval algorithm that yields the original E-field vector time-domain waveform of the elliptically-polarized terahertz pulse.

Finally, we stress that our formalism is general and it does not restrict the frequency bandwidth of the measurement, because it allows for choosing any proper EO crystal. Therefore, polarization spectroscopy by PS-EO sampling is applicable not only in the terahertz frequency region but also in mid- and near-infrared frequency regions, and thus a variety of applications can be expected in the future.

**Funding:** This work was partially supported by Japan Science and Technology Agency (JST) under Collaborative Research Based on Industrial Demand "Terahertz-wave: Towards Innovative Development of Terahertz-wave Technologies and Applications", and Japan Society for the Promotion of Science (JSPS) KAKENHI (JP17J04085, JP18H02040).

**Conflicts of Interest:** The authors declare no conflicts of interest.

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
