# Peer review of "Polarization-Sensitive Electro-Optic Sampling of Elliptically-Polarized Terahertz Pulses: Theoretical Description and Experimental Demonstration"

_2571-712X, doi:10.3390/particles2010006_

Round 1

Reviewer 1 Report

Dear Authors,

The manuscript reviews authors’ very recent work on the polarization-sensitive electro-optic (PS-EO) sampling methods used to characterize an elliptically-polarized terahertz (THz) pulses. Accurate characterization of an elliptically-polarized (including circularly-polarized) THz electromagnetic waveform plays a vital role in spectroscopic characterization of various materials such as some new artificial chiral materials and biological samples and is in an urgent need for the study of new materials in the THz frequency range. Therefore, the topic of the manuscript holds important impact in the THz community.

I particularly like the way that authors used to introduce the basic concept for their PS-EO method: Authors start with the very basic and ideal case for EO sampling of linearly-polarized THz pulses without considering velocity mismatch, and then take into account different effects such as phase-mismatch, absorption, and etc one by one. At the end, authors use very convincing experimental results to verify their theoretical formulation and the validity of the approach. Even though I am not a very good theorist, I could understand their model and the algorithm pretty well.

The manuscript is very well written in terms of English language in an extremely good logic. Throughout the entire manuscript, I can only find one typo on line 286: “… to assume ae delta-function-like …” should be ““… to assume a delta-function-like …”.

Out of curiosity, may I ask a question, can authors’ approach (maybe after some modification) be applied to PS-EO sampling of an arbitrarily-polarized THz pulse?

In conclusion, I would strongly suggest publication of the manuscript on Particles as is.

Sincerely yours,

Reviewer

Author Response

Please check the attached PDF file.

Reviewer 2 Report

The authors reviewed their progress in the polarization-sensitive electro-optic sampling of elliptically-polarized terahertz pulses. By considering the velocity mismatch between the probe beam and terahertz wave, they retrieved the terahertz electric field based on the multilayer model in frequency domain description. They experimentally confirmed that the measured signals are independent with the electro-optic crystals which are used for detection. The manuscript is technically sound and the claims are supported by the presented data. Before I recommend this manuscript for publication in Particles, I have the following comments:

1. The conclusion (1) in the abstract should be modified, which is “the polarization state of each frequency component can be obtained by simply comparing relative amplitude and phase of two mutually orthogonal directions”. I think the description in this way is only a basic fact, which cannot be used as a conclusion for the authors. I think the author intended to claim that the amplitude and phase can be obtained simultaneously, which can be used for calculating the polarization for each frequency directly. 

2. The sentence in the abstract (“We experimentally demonstrate that the obtained polarization-dependent terahertz spectra and the retrieved E-field vector time-domain are independent of the EO crystal used for detection.”) is a little ambiguous and wordy. I think the authors mainly intend to state the signal is independent of the crystals for detection. Because the certain waveform indicates the certain spectra after Fourier Transform, which is an obvious fact. 

3. The authors claim that the velocity mismatch between the probe beam and the terahertz signal plays a role in measurements. I think it will be better to quantify the effect of the mismatch, especially in the very beginning of the manuscript, such as in the introduction section. The authors can compare the signal with and without considering the velocity mismatch. Or please describe the effect more concise in the manuscript

4. It will be better if the authors discuss the chirality as well as the polarization in the manuscript. It will be very useful for the reader who is concerning this. This will also help generalize this technique in biology and chemistry applications. 

5. The bridge between “Pockels effect” and “Frequency domain description” is built by the Fourier Transform. Both of them are related to the nonlinear effect χ(2), which describe the same phenomenon in different basis. It should be clearly stated in the manuscript. 

6. Some of the sentences are wordy. Hope the manuscript can be modified in a simple and concise way. Please use the same tense (past or present) in the whole manuscript.   

Author Response

Please check the attached PDF file.

Round 2

Reviewer 2 Report

All the comments raised in the previous review are addressed well in the current version. 

Particles EISSN 2571-712X Published by MDPI AG, Basel, Switzerland RSS E-Mail Table of Contents Alert
Back to Top